# A Liquid Chromatography-Mass Spectrometry Method to Study the Interaction between Membrane Proteins and Low-Molecular-Weight Compound Mixtures

**DOI:** 10.3390/molecules27154889

**Published:** 2022-07-30

**Authors:** Hideo Ogiso, Ryoji Suno, Takuya Kobayashi, Masashi Kawami, Mikihisa Takano, Masaru Ogasawara

**Affiliations:** 1Toyama Prefectural Institute for Pharmaceutical Research, Imizu 939-0363, Toyama, Japan; masaru.ogasawara@pref.toyama.lg.jp; 2Department of Medical Chemistry, Kansai Medical University, Hirakata 573-1010, Osaka, Japan; sunory@hirakata.kmu.ac.jp (R.S.); kobayatk@hirakata.kmu.ac.jp (T.K.); 3Department of Pharmaceutics and Therapeutics, Graduate School of Biomedical and Health Sciences, Hiroshima University, Hiroshima City 734-8553, Hiroshima, Japan; ma-kawami@hiroshima-u.ac.jp (M.K.); takanom@hiroshima-u.ac.jp (M.T.)

**Keywords:** molecular interaction, high-resolution mass spectrometry, size-exclusion chromatography, membrane protein, ligand-receptor complex, GPCR

## Abstract

Molecular interaction analysis is an essential technique for the study of biomolecular functions and the development of new drugs. Most current methods generally require manipulation to immobilize or label molecules, and require advance identification of at least one of the two molecules in the reaction. In this study, we succeeded in detecting the interaction of low-molecular-weight (LMW) compounds with a membrane protein mixture derived from cultured cells expressing target membrane proteins by using the size exclusion chromatography-mass spectrometry (SEC-MS) method under the condition of 0.001% lauryl maltose neopentyl glycol as detergent and atmospheric pressure chemical ionization. This method allowed us to analyze the interaction of a mixture of medicinal herbal ingredients with a mixture of membrane proteins to identify the two interacting ingredients. As it does not require specialized equipment (e.g., a two-dimensional liquid chromatography system), this SEC-MS method enables the analysis of interactions between LMW compounds and relatively high-expressed membrane proteins without immobilization or derivatization of the molecules.

## 1. Introduction

Biological macromolecules interact with other biomolecules or bioactive low-molecular-weight (LMW) compounds via covalent bonds, or noncovalent interactions, such as hydrogen bond, electrostatic, and hydrophobic interactions. Consequently, a specific function is expressed through conformational changes in biomolecules. Therefore, elucidating the specific interactions between these molecules will lead to a better understanding of the biological phenomena at the molecular level. Several methods for molecular interaction analysis have been widely used to investigate the mechanisms underlying the action of various bioactive substances and to develop new drugs [1,2,3]. During interaction analysis of biomolecules and bioactive substances, receptor proteins or ligand molecules are generally immobilized or labeled to facilitate detection of the binding process when mixed in a solution. This is often detected by physicochemical methods, such as surface plasmon resonance (SPR) and quartz crystal microbalance [4,5]. There are two major limitations to these existing methods. (1) Immobilization or labeling of the molecules is necessary. The possibility that chemical modifications may affect the intermolecular interactions must be considered. (2) It is necessary to prepare purified samples of receptors or ligand molecules and one or both target molecules must be identified before binding can be analyzed. In addition, although isothermal titration calorimetry does not require labeling or immobilization, it does require relatively high concentrations and large amounts of purified samples [6]. Therefore, when multiple components in small amounts contribute to physiological effects, such as metabolites in vivo, the analysis of the interaction in complex systems is currently highly labor-intensive. Since many membrane proteins, such as receptor tyrosine kinases, ion pumps, and G-protein coupled receptors (GPCRs), have been reported as target receptors for drugs, it is desirable to develop a method of interaction analysis for membrane proteins [2,7,8]. However, the number of interaction analysis methods that can be applied to ligand screening is limited because of the difficulty in handling membrane proteins.

Another method is size-exclusion chromatography (SEC), which can separate proteins complexed with their low molecule ligands by size and have been used to determine whether complexes are formed [9,10,11,12]. If the LMW ligand is fluorescent, the interaction can be analyzed with relative ease using SEC; if not, it is necessary to use techniques to specifically detect the complexes. When using mass spectrometry to identify bound molecules, SEC, which separates native complexes, can be used as the inlet side of the mass spectrometer; however, care must be taken (e.g., using volatile salts in place of non-volatile salts). Mass spectrometry for native proteins is generally less sensitive than the detection sensitivity of LMW compounds. For membrane proteins, detergent additives are required, which unfortunately induce ion suppression and increase the difficulty of sensitive analysis.

To date, three different approaches have been utilized for mass spectrometry-based ligand screening analysis of interactions between proteins and LMW ligands. (1) LC-MS measurements for each fraction (conventional LC-MS), during which the ligand complex is fractionated by ultrafiltration or affinity gel and then the binding ligand is extracted followed by LC-MS measurement [13,14,15]. Although step-wise, it has the advantage that the LMW ligands can be measured with high sensitivity. On the other hand, no information on the protein side can be obtained. Robotic automation is needed to increase throughput. (2) Two dimensional LC-MS (2D-LC-MS), which separates ligand-protein complexes by LC such as SEC or affinity chromatography and then automatically introduces each eluate into a second LC to measure sub-molecules derived from the complexes (e.g., the dissociated ligand; [16,17,18,19]. The elution profile in the first dimensional LC provides information on the complex and the second dimensional LC has the advantage that the ligand is measured with high sensitivity. Special 2D-LC systems need to be constructed. (3) The native mass spectrometry method (native MS) is used to measure the mass increase in complex molecules formed as a result of intermolecular interactions [10,11,20,21,22,23,24,25]. Native MS, the latest method, provides information on the protein side, stoichiometry of ligand binding, and concomitant binding of different ligands. There are issues such as the inclusion of non-specific binding and the fact that it is not always highly sensitive. The methods described above complement each other. While recent attempts have been made to study molecular interactions in silico [26,27], experimental investigations are becoming increasingly significant.

In this study, we developed a method to assess ligands bound to membrane proteins using only one-dimensional SEC as a separation tool. To achieve this, it was necessary to overcome the issue of ion suppression caused by detergents and relatively high salt concentrations in the mobile phase, in addition to large quantities of other components co-eluting during the SEC. This led to the establishment of a simple method for the evaluation of bound ligands without a two-dimensional LC system. This method offers an alternative for interaction analysis. Moreover, it will also be useful as a simplified approach for estimating and screening molecules involved in interactions between LMW compound mixtures and membrane protein mixtures, without requiring advance identification of the LMW ligands or proteins involved in the interaction. For example, by analyzing the interactions between a drug mixture with a membrane protein mixture obtained from cancer cells derived from patients who exhibit acquired resistance to anticancer drugs, it will be possible to screen for effective drugs without having to identify in advance the proteins responsible for drug resistance, such as receptors and enzymes.

## 2. Results

### 2.1. Development of a SEC-MS Method for Detecting Interactions between A431 Cell-Derived Membrane Proteins and Lapatinib 

To develop a method for the analysis of interactions between a membrane protein mixture and LMW compound mixture, we chose model systems in which the target proteins were highly expressed in cultured cells that interact with a specific ligand. As a first workbench for an optimization example, we extracted membrane proteins from A431 cells with a high expression of epidermal growth factor receptor (EGFR) and attempted to detect the interaction between A431 cell-derived membrane proteins and lapatinib, a synthetic inhibitor of EGFR and HER2. 

As mobile phase conditions for SEC are directly linked to mass spectrometry, the species and concentrations of salts and detergents were optimized so as not to interfere with mass spectrometry measurements. We chose detergents, such as lauryl maltose neopentyl glycol (LMNG) and *N*-dodecyl-β-d-maltoside (DDM), to solubilize the membrane proteins from cultured cells under non-denaturing conditions [25,28,29]. The resulting mobile phase for the SEC was 100 mM ammonium formate (pH 7.4), containing 0.001% LMNG and 5% acetonitrile. LMNG was added to the mobile phase at slightly above the critical micelle concentration (CMC) to maintain the solubilized state of the membrane proteins without affecting the ionization of LMW compounds. A low concentration of acetonitrile in the mobile phase was effective in reducing the accumulation of lapatinib in the flow path during repeated measurements. In addition, the composition of the post-column solution, which was combined immediately before ionization, was optimized. These ionization conditions were important for the dissociation and efficient ionization of LMW ligands in the complex. It was absolutely needed to avoid ionization suppression by large quantities of other components. Thus, instead of the electrospray ionization (ESI) mode, the atmospheric pressure chemical ionization (APCI) mode, which is less prone to ion suppression, was adopted. In practice, the lapatinib signal was not detected using the ESI mode in this SEC-MS system. Moreover, the washing conditions of the SEC column between sample measurements were optimized. These washing conditions were important for reproducible signal detection in repeated measurements. As a result, we were able to reproducibly analyze the interactions between lapatinib and membrane proteins. See the Materials and Methods, Section 4 of this report for details. The elution profile of free lapatinib after dissociation from the protein complex was monitored using *m*/*z* 581.143 (Figure 1). At the same time, monitoring the signals in the *m*/*z* 533–1019 range, excluding the saturated signals of ions in the entire MS spectrum, gave a chromatogram roughly corresponding to the elution profile of the proteins monitored at 280 nm absorption. These ions may originate from a group of LMW compounds such as detergents that bind nonspecifically to membrane proteins, while they are all monovalent ions because of the APCI.

When A431 cell-derived membrane protein fractions of overexpressed EGFR were mixed with lapatinib, an elution peak of lapatinib was observed on the higher molecular weight side before a retention time of 11.5 min in the SEC chromatogram (Figure 1A). The small molecule group was eluted at ~12 min. To prevent contamination of the mass spectrometer, it flowed into the waste line so that it was not introduced into the mass spectrometer after 11.5 min. When lapatinib was mixed with a membrane protein fraction derived from CHO (Chinese hamster ovary) cells, which do not express EGFR, instead of A431 cells, no elution peak for lapatinib was observed. The elution time of lapatinib was approximately 9.1 min, which is lower than the elution time of BSA (approximately 10.1 min), suggesting that it existed as a complex of more than 70 kDa. This is consistent with the interpretation of lapatinib bound to EGFR, which had a molecular weight of approximately 180 kD. When the membrane fraction was trypsinized or heat-treated in advance, interaction with lapatinib was no longer detected (Appendix A), indicating that lapatinib bound to some proteins. Thereafter, the concentration-dependent effect of lapatinib was examined using this method. The results showed that the 50% binding concentration value (C50) was approximately 100 nM, which is approximately 10 times higher than the IC50 (10 nM) of lapatinib [30]. This is reasonable considering the fact that it was diluted by SEC during the measurement (Figure 1B,C). When lapatinib was added to the membrane protein fraction together with an excess of ATPγS, lapatinib binding was inhibited (Figure 1D). This finding is consistent with lapatinib being a competitive inhibitor of ATP [31], while ATPγS was not detected by APCI and its binding to EGFR could not be confirmed. These results indicate that the interaction between the membrane proteins, EGFR, and its specific ligand, lapatinib, can be analyzed using this SEC-MS method without immobilization, labeling, or protein purification.

### 2.2. Detection of MiaPaCa Cell-Derived Membrane Protein Interactions with LMW Inhibitors

As the second workbench, we attempted to detect the interaction between a GPCR and its ligand. As an example, we chose a combination of MiaPaCa cells expressing CXCR4 and its antagonist WZ811; however, contrary to our expectations, these interactions were not detected. We hypothesized that this was because of the low expression of CXCR4, and performed proteomic analysis of membrane proteins in MiaPaCa cells to select the highly expressed protein species. Using conventional LC-MS, and not nano-LC-MS with high sensitivity, only major membrane proteins were identified in the membrane fraction from MiaPaCa cells. MASCOT analysis showed that the most abundant membrane protein was mitochondrial ATP synthase, based on the exponentially modified protein abundance index (emPAI) parameter as an index of abundance [32] (Appendix A). We detected an interaction between ATP synthase with an emPAI of 2.2 and its inhibitor, oligomycin, indicating that the C50 was approximately 200 nM (Figure 2A,B). We also tried to analyze the combination of lapatinib and EGFR with an emPAI of 0.2 in the membrane protein fraction from MiaPaCa2, and this interaction was detected at slightly above the detection limit (Figure 2C). These verification results using the A431 and MiaPaCa2 cells indicate that our SEC-MS method can be used for ligand binding analysis to highly expressed membrane proteins.

### 2.3. Interaction of Cultured Cell-Derived Membrane Proteins with Ingredients Extracted from Paeonia lactiflora

As an example of the application of this method, we attempted to clarify the interaction between a membrane protein mixture and LMW compound mixture using the method verified above. As an example of an LMW compound, we examined ingredients extracted from *P. lactiflora*. The dry roots of *P. lactiflora* have been used for more than 1000 years to treat various diseases [33,34,35]. The various ingredients contained in herbal medicines may act in concert with each other. As an example of membrane proteins, macrophage-like cell J774-derived membrane proteins were used to analyze the interactions with peony ingredients. Using separation by reversed-phase HPLC (RPLC) and positive ionization by APCI, we have already determined the *m*/*z* signals corresponding to the ingredients in a previous study [36] (Appendix A). Focusing on the top 50 components among all ingredients detected through RPLC analysis, the interactions were evaluated from mass chromatograms of the *m*/*z* signals of the individual components on the SEC-MS measurement (Figure 3). Relatively clear chromatograms of the bound compounds were obtained in the negative ion mode. The range of *m*/*z* 558–1019 was monitored in the case of the negative ion mode in order to obtain the overall elution profile. Two components were found to interact with some proteins. These were C_15_H_14_O_6_, identified as catechin or epicatechin, and C_15_H_12_O_6_, previously deduced as okanin from the MS/MS spectrum. Since these interactions were also detected in CHO-derived proteins, they were determined to be interactions involving common proteins rather than J774-specific proteins.

### 2.4. Detection of the Interaction of Human M2 Muscarinic Acetylcholine Receptor-Overexpressing Sf9 Cell-Derived Membrane Proteins with Atropine

Using Sf9 cells overexpressing GPCR, which is a human M2 muscarinic acetylcholine receptor (M2R) [37], we attempted to detect the interaction between GPCR and its ligand; however, contrary to our expectations, these interactions could not be detected using the SEC-MS method verified above. Having identified the factors that hinder detection, it is now possible to detect the ligand binding to protein by modifying the SEC conditions (see Materials and Methods, Section 4). Notably, the atropine-M2R complex seemed to be unstable during passage through the SEC column. Thus, shortening the column length to 10 mm and reducing the column temperature to 5 °C enabled the detection of ligands bound to GPCRs, although separation among proteins by molecular size was no longer expected (Figure 4A). The small molecule group was eluted at ~2.5 min and not introduced into the mass spectrometer. The elution peak of atropine binding to protein was observed around 1.5 min. Using a membrane protein fraction derived from control Sf9 cells that did not express M2R, the 1.5 min peak for atropine was not observed. This indicates that the 1.5 min peak corresponds to atropine bound to M2R. Thereafter, the concentration-dependent effect of atropine was examined using the modified SEC-MS method (Figure 4B). To stabilize the structure of GPCRs, they are generally solubilized and purified with addition of ligands. In the present study, two patterns were analyzed using membrane protein fractions prepared with or without atropine added during solubilization (Figure 4A, B; Appendix A). The resultant 50% binding concentration value (C50) was approximately 700 nM, which is approximately 200 times higher than the IC50 (4 nM) of atropine [30]. Simultaneous addition of excess tiquidium, a different M2R antagonist, to the membrane protein fraction inhibited atropine binding, suggesting competitive inhibition (Figure 4C). As with ATPγS, owing to the low ionization efficiency of tiqudium by APCI, the interacting tiquidium could not be directly detected. The fact that the atropine signal around 1.5 min, which was only detected in M2R-expressing Sf9 cells and not in control Sf9 cells, was found to be suppressed by the same M2R antagonist, confirmed that this SEC-MS method allows for the unstable but specific interaction of M2R and atropine to be detected.

## 3. Discussion

The 2D-LC-MS method is used to analyze interactions between proteins and LMW ligands. In 2D-LC-MS, SEC is performed in the first dimension and then the SEC eluate is introduced into a second LC, such as an RP-LC, using a programmed switching valve to analyze LMW compounds. This system is superior in that the SEC eluate is not introduced directly into the mass spectrometer, and so mobile phases containing non-volatile salts and detergents can be used, and LMW compounds can be measured with high sensitivity. However, to perform this, a special system needs to be constructed. If not, this operation is carried out offline. This means that samples fractionated by SEC need to be measured by LC-MS one-by-one, which is labor-intensive and time-consuming. Therefore, a method to directly link SEC to a mass spectrometer was developed in this study. To achieve this, salts in the mobile phase should be volatile, and detergents should be non-ionic with low CMC so as not to interfere with ionization. In addition, the eluate of SEC should be acidified in-line so that dissociated ligands are specifically detected by mass spectrometry. It was possible to avoid ion suppression as much as possible by using the 0.001% LMNG-containing mobile phase and APCI, which set the conditions for analysis that could reproducibly detect trace amounts of ligand bound to membrane proteins in the presence of miscellaneous background signals. As a result, we developed a method for analyzing the interaction between a membrane protein mixture and an LMW compound mixture using SEC-MS. This SEC-MS method was applied and verified for the analysis of the molecular interactions between specific inhibitors, such as lapatinib and oligomycin, and their target membrane proteins (Figure 1 and Figure 2). This novel simpler method, as well as the other popular methods, facilitate evaluation of the capacity of ligands to bind to target membrane proteins and analysis of competitive inhibition. The advantage of this SEC-MS method is that it allows interaction analysis without the need for immobilization, labelling, purification, and isolation of specific molecules of both the ligand and receptor.

While we tried to determine whether interactions could be detected using this method for drugs that target GPCRs, the interactions could only be detected using the modified condition of the SEC-MS method (Figure 4). It is generally known that GPCRs become unstable after solubilization, and the atropine-M2R complex herein became unstable as it passed through the SEC column. The SEC-MS method developed in this study was not well suited for ligand binding analysis of GPCRs, as binding of the ligand to M2R was only detected by reducing the column transit time at the expense of separation and by lowering the temperature. The fact that the 50% binding concentration of 700 nM in this study was more than 200-fold higher than the IC50, despite sacrificing SEC separation, could be interpreted as the operating conditions destabilizing the M2R-atropine complex. Thus, further improvements are need for GPCR analyses.

The medicinal effects of Chinese herbal formulas containing various ingredients are believed to be expressed through the coordinated action of multiple ingredients [38,39,40,41]. In most cases, the action of a single ingredient is relatively mild; therefore, unknown complex actions are presumed to be involved in the expression of medicinal effects. Our SEC-MS method enables analysis of intermolecular interactions without the requirement of specifying the target, and thus facilitates mixture-to-mixture interactions. We used this method to find LMW compounds that interact with cell membrane proteins among all the ingredients of a herb. As an example, we examined the interaction of a mixture of components extracted from *P. lactiflora* with a mixture of membrane proteins solubilized from cultured cells, J774, or CHO. The results showed that the two components interacted with some of the proteins common to the two types of cells. Based on the mass chromatograms of these components, it was inferred that they interacted with multiple proteins rather than binding to a specific protein (Figure 3). Based on the peak shapes and signal intensities corresponding to the interactions in this measurement, we estimate that a significant number of non-specific interactions are also included. In the future, we plan to identify herbal medicine components that specifically interact with immune cells. Other plans include investigating which in-gut metabolites interacts with immune cells.

Possible applications of this SEC-MS method include the following. If a mixture of LMW compounds affects a particular cell, the method can be used as a clue to determine which compounds are involved in the biological activity. It can also be used as a means of identifying target membrane proteins when combined with gene knockdown and other methods. We anticipate that this SEC-MS method will be used to search for off-targets for specific drugs. Although this method is less sensitive than 2D-LC-MS, it is more convenient and throughput-oriented and provides information on the approximate molecular weight of the receptor to which the ligand binds. For more sensitive ligand detection, we also used conventional LC-MS in combination with off-line fractionation. Finally, we are also investigating the possibility of using ion-exchange chromatography, which is different from SEC, as a separation mode for membrane protein complexes on the inlet side.

### Limitations of the Study

This analysis was technically challenging, and the method is limited to non-covalent interactions with highly expressed and stable membrane proteins. Further improvements are needed to detect a wide range of interactions with membrane proteins and GPCRs that are expressed in small amounts. In this method, APCI, which is less susceptible to ionization suppression than ESI, is used as an ionization method to sensitively detect specific compounds in the presence of many other compounds and proteins. For this reason, in addition to the ATPγS and tiquidium used in this study, compounds such as nucleic acid-related compounds that are not detected by APCI cannot be analyzed. Unfortunately, if the *m*/*z* signal, which specifically detects LMW compounds, overlaps with a background signal, it impacts on the measurement.

## 4. Materials and Methods

### 4.1. Key Materials

Key materials used in this study were listed in Appendix A.

### 4.2. Cell Culture

Cells were cultured in 10 mL medium supplemented with 10% FBS, penicillin (100 U/mL), and streptomycin (100 μg/mL) in a 100 mm dish. Depending on the cell line, DMEM for A431 and MIA PaCa2, and Ham’s F12 for CHO-K1 were used. Media were changed two or three times a week until the cells reached 60% to 80% confluency.

### 4.3. Method Details

The outline of the method was shown in Appendix A.

#### 4.3.1. Preparation of Membrane Protein-Enriched Fractions

Cultured cells that reached 60% to 80% confluency were washed with PBS and then scraped with 1 mL of hypotonic buffer (20 mM HEPES-NaOH buffer pH 7.4, protease inhibitor cocktail, cOmplete (Roche, Mannheim, Germany)). The suspension was centrifuged at 14,000× *g* for 10 min at 4 °C to obtain cell pellets. The cells were suspended in 0.5 mL isotonic buffer (20 mM HEPES-NaOH buffer pH 7.4, 150 mM NaCl, protease inhibitor cocktail, cOmplete) and were sheared by passing the suspension through an 18-gauge needle 10 times, followed by centrifugation at 14,000× *g* for 10 min at 4 °C. The resulting pellet was resuspended in 0.45 mL isotonic buffer. Thereafter, 0.05 mL of 1% (*w*/*v*) LMNG aqueous solution and 0.025 mL of 10% (*w*/*v*) DDM aqueous solution were added to the suspension and the microtube containing the sample was rotated for 1 h at 4 °C to solubilize membrane proteins. After further centrifugation at 14,000× *g* for 10 min at 4 °C, the supernatant was obtained, and 600 mg of Bio-Beads SM-2 with 0.5 mL of 20 mM HEPES-NaOH buffer pH 7.4 were added, followed by rotation for 1 h at 4 °C. After excess detergent was adsorbed onto the Bio-Beads SM-2 resin, it was removed by filtration, and the membrane protein solution was concentrated to 0.1 mL using an Amicon Ultra (10 kDa cutoff, Millipore, Billerica, MA, USA) centrifugal filter, followed by the addition of 0.4 mL of 20 mM HEPES-NaOH buffer pH 7.4. This ultrafiltration operation was repeated twice, and the solution was concentrated to 0.05 mL. Protein concentration in the presence of 0.1% (*w*/*v*) SDS was measured using the FluoProdige Assay Kit (OZ Biosciences, San Diego, USA) and was typically 15–20 mg/mL. The membrane protein fraction was stored in 10 μL aliquots at −80 °C until use.

#### 4.3.2. Ligand-Receptor Complex Formation

Typically, 1–5 μL of LMW compounds dissolved in organic solvent were dried using a centrifugal evaporator (TAITEC Spin Dryer Lite VC-36R, EYELA UT-1000, Koshigaya, Japan), and redissolved in a mixture of 5 μL of the membrane protein fraction and 5 μL of the ligand binding buffer [20 mM HEPES-NaOH pH 7.4, 300 mM NaCl, 0.05% (*w*/*v*) LMNG, and 0.05% (*w*/*v*) DDM]. After incubation for 10 min on ice, the solution was centrifuged at 14,000× g for 10 min at 4 °C. The supernatant (9 μL) was prepared for SEC-MS analysis.

#### 4.3.3. Liquid Chromatography

SEC-MS was performed using a Shimadzu LC20A UHPLC system (Kyoto, Japan) coupled with a Bruker maXis II TOF mass spectrometer (LC-TOF/MS, Bremen, Germany). Typically, 4 μL of sample was injected, and components were separated through a SEC column, Inertsil WP300 Diol (100 × 2.1 mm, GL Sciences, Tokyo, Japan) with guard column (10 × 1.5 mm, GL Sciences, Tokyo, Japan), using mobile phase A [100 mM ammonium formate, pH 7.4, 0.001% (*w*/*v*) LMNG, and 5% (*v*/*v*) acetonitrile]. Flow rate and column temperature were set to 0.025 mL/min and 20 °C. After mobile phase B (ultra-pure water/methanol/chloroform/formic acid (80:120:1:0.5)) was mixed post-column at 0.175 mL/min, the eluent was introduced into the mass spectrometer. To avoid contaminating the mass spectrometer, the eluate was introduced into the mass spectrometer only for the higher-molecular fraction with 4–11.5 min of elution time. The duration of each measurement was 20 min. To ensure reproducible SEC-MS measurements, the following conditioning steps were performed prior to each sample injection. After injecting 40 μL of washing solution 1 (1 M sodium chloride/0.25 M EDTA/1% (*w*/*v*) LMNG/10% (*w*/*v*) DDM (79.2:9.9:9.9:1.0, *v*/*v*/*v*/*v*), i.e., the mixing ratio of 80:10:10:1), mobile phase A was run at 0.1 mL/min for 5 min, then 40 μL of washing solution 2 (70% (*v*/*v*) acetonitrile) was injected and the mobile phase was run in the same manner. The solution 2-wash was repeated three times, and 40 μL of washing solution 1 was injected again; the mobile phase flowed in the same manner. During conditioning, the valve was switched to a waste line to avoid the contamination of the mass spectrometer. The time required for one cycle was 45 min, including the runtime of the cleaning and measurement.

#### 4.3.4. Mass Spectrometry

To reduce ionization suppression effects as much as possible, an APCI source was used instead of ESI. The capillary voltage was set to 3000 V in the positive ion mode and −2000 V in the negative ion mode. The corona current was set to 4000 nA in the positive-ion mode and 9000 nA in the negative-ion mode. The vaporizer temperature was set to 350 °C. Other parameters were set according to the manufacturer’s recommendations. The LC-TOF/MS system was controlled using the otofControl and Hystar software (Bruker, Bremen, Germany). LMW compounds were measured by a full scan of the parent ion (MS1) within *m*/*z* 50–2200. Before sample measurements, the *m*/*z* values were calibrated using a calibrant (APCI-L Low Concentration Tuning Mix, Agilent, Santa Clare, CA, USA) so that the mass accuracy was within 5 ppm. The absolute threshold was set to 250 cts in positive ion mode and 286 cts in negative ion mode to eliminate the noise signal. As the Bruker maXis II TOF can measure mass to within 2 ppm accuracy, mass chromatograms of LMW compounds were drawn with a width of ±0.005.

#### 4.3.5. Profiling Membrane Proteins

To determine the species and amount of membrane proteins in the membrane protein fraction prepared from the MIA PaCa-2 cells, proteomic analysis was performed using the conventional LC-TOF/MS system described above. Peptide mapping was performed using a modified version of a previously published method [42]. Under this approach, 5 μL of the membrane protein fraction described above was mixed with 5 μL of PBS, 33 μL of guanidinium chloride, and 1.5 μL of 0.1 M dithiothreitol. It was then incubated at 37 °C for 30 min to denature and reduce proteins. Thereafter, 3.5 μL of 100 mM iodoacetamide was added and incubated at room temperature for 15 min for alkylation of the thiols. Using a Zeba Spin Desalting Column (Thermo Fisher Scientific, Waltham, MA, USA), the buffer was exchanged with 100 mM ammonium bicarbonate. The resultant solution (70 μL) was mixed with 7 μL of acetonitrile and 5 μL of trypsin (1 mg/mL) and incubated at 37 °C overnight for digestion. The resultant solution was stored at −80 °C until use.

Thereafter, 16 μL of sample was injected, and peptides were separated through an AQUITY UPLC BEH C18 column (1.0 × 100 mm, 1.7 µm, Waters, Milford, CT, USA), using mobile phase A (0.1% (*v*/*v*) formic acid) and mobile phase B (0.1% (*v*/*v*) formic acid in acetonitrile). The gradient was changed from 98/2 (A/B) to 93/7 (A/B) over 0.5 min, and then to 92/8 (A/B) over 0.5 min, to 80/20 (A/B) over 29 min, to 60/40 (A/B) over 20 min, to 2/98 (A/B) over 1 min, held for 7 min, changed to 98/2 (A/B) over 1 min, and held for 6 min for re-equilibration. The flow rate and column temperature were set at 0.1 mL/min and 55 °C. The duration of each measurement was 65 min. The calibrant (ESI-L Low-Concentration Tuning Mix, Agilent, Santa Clare, CA, USA) was introduced into the mass spectrometer at 52–53 min. For peptide mapping analysis, ESI was used in the positive ion mode. The capillary voltage was set to 4500 V. The vaporizer temperature was set to 350 °C. Peptides were measured by a full scan of the parent ion (MS1) within *m*/*z* 100–3000. Collision induced dissociation (CID)-MS/MS (MS2) measurements were performed using a data-dependent auto MS/MS mode at a spectral rate of 5–20 Hz. Other parameters were set according to the manufacturer’s recommendations. Each measurement data point was recalibrated using the DataAnalysis software (v.4.4SR1, Bruker, Bremen, Germany) and then converted to .mgf files for MASCOT analysis. A peak list in the .mgf file was searched with MASCOT v.2.7 (Matrix Science, London, UK) using the following parameters: taxonomy = Homo sapience; enzyme = trypsin; maximum missed cleavages = 2; fixed modifications = carbamidomethylation of cysteine; variable modifications = oxidation of methionine; peptide mass tolerance = 0.02 Da, MS/MS tolerance = 0.02 Da. We selected membrane proteins from the protein list identified by the MASCOT search and determined that these were the major membrane proteins in the membrane protein fraction (Appendix A).

#### 4.3.6. Profiling Peony Ingredients

To determine the species and amount of ingredients in peony, LMW compounds were analyzed using the conventional LC-TOF/MS system described above. A total of 10 mg of peony powder, made from dry roots of *P. lactiflora*, was suspended in 0.5 mL of methanol, and mixed for 5 min followed by evaporation using a centrifugal evaporator at 60 °C for 30 min. Then, 700 μL of water-saturated 1-butanol (BuOH), 0.15 mL ultra-pure water, and 0.05 mL ammonium acetate buffer pH 5.8 were added to the dry materials. After vigorous shaking for 5 min and sonication for 1 min in an ultrasonic bath (Branson 5510, St. Louis, MO, USA), the samples were centrifuged at 10,000× *g* for 5 min, after which, 0.6 mL of the upper layer was collected in a 2 mL polypropylene vial. The original suspension was re-extracted by adding BuOH (0.7 mL), followed by centrifugation. The second BuOH extract (0.7 mL) was added to the first extract. After the addition of 0.5 mL of methanol, 0.1 mL of the solution was evaporated using a centrifugal evaporator at 40 °C for 60 min and then re-dissolved in 0.5 mL of LC mobile phase A (described below). This solution was finally filtrated using a 0.45 μm spin filter (Ultrafree-MC-HV).

The ingredients were separated on a Unison UK-C18 column (100 × 2 mm, 3 μm, Imtakt, Kyoto, Japan). Mobile phase A was isopropanol/methanol/water (5:1:4) with 0.2% (*v*/*v*) formic acid and 0.028% ammonia. Mobile phase B was isopropanol, 0.2% (*v*/*v*) formic acid, and 0.028% ammonia. The gradient conditions were as follows: change from 100/0 (A/B) to 50/50 (A/B) over 3 min, and then to 20/80 (A/B) over 6 min, hold for 2 min, change to 100/0 (A/B) over 1 min, and then hold for 12 min for re-equilibration. The flow rate was 0.1 mL/min, and chromatography was performed at 40 °C. Typically, 2 µL of the sample was injected. For ingredient analysis, APCI was used in the negative ion mode. The duration of each measurement was 24 min. The calibrant (APCI-L Low-Concentration Tuning Mix) was introduced into the mass spectrometer from 0 to 5 min. In mass spectrometry measurements, capillary voltage was set to −2000 V, corona current was set to 9000 nA in negative ion mode, and the vaporizer temperature was set to 350 °C. Ingredients were measured by a full scan of the parent ion (MS1) within *m*/*z* 50–2200. CID-MS/MS (MS2) measurements were performed in data-dependent auto MS/MS mode at a spectral rate of 1 Hz. Other parameters were set according to the manufacturer’s recommendations.

Each measurement data was processed using the MetaboScape software (v.4.0, Bruker, Bremen, Germany). The components were identified from the elution peaks, and their exact mass values, and the molecular formulae were calculated. Searching the database of ingredients in peony and other herbal medicines, ingredient names were speculated based on the molecular formula and MS/MS spectrum.

#### 4.3.7. Method Modification for GPCR (Sf9 Cells Overexpressing M2R)

Sf9 cells overexpressing M2R (M2-BRIL fusion protein; S110R mutant) were obtained as previously described [38]. The cell suspension with PBS (approximately 50% slurry) was stored at −80 °C until use. A total of 50 μL of the suspension was diluted in 1 mL of isotonic buffer (described above) and was centrifuged at 14,000× *g* for 10 min at 4 °C. The resultant pellet was resuspended in isotonic buffer (0.5 mL), and the cells were sheared by passing the suspension through an 18-gauge needle 10 times. The suspension was centrifuged again, and the resultant pellet was resuspended with 0.45 mL of isotonic buffer. Thereafter, 10 μg of atropine in 10 μL of methanol solution was added to the microtubes and dried. Thereafter, 0.45 mL of the cell disruption solution was mixed to stabilize M2R. Thereafter, 0.05 mL of 1% (*w*/*v*) LMNG aqueous solution, 0.05 mL of the 10% (*w*/*v*) DDM, and 1% (*w*/*v*) cholesteryl hemisuccinate (CHS) aqueous solution mixture were added to the suspension, and the microtube containing the sample was rotated for 1 h at 4 °C to solubilize membrane proteins. After centrifugation at 14,000× *g* for 10 min at 4 °C, the supernatant was obtained, and then 1200 mg of Bio-Beads SM-2 with 0.3 mL of 20 mM HEPES-NaOH buffer pH 7.4 was added, followed by rotation for 1 h at 4 °C. After excess detergent was adsorbed onto the Bio-Beads SM-2 resin, they were removed by filtration, and the membrane protein solution was concentrated to 0.1 mL using an Amicon Ultra (10 kDa cutoff) centrifugal filter, followed by the addition of 0.4 mL of 20 mM HEPES-NaOH buffer pH 7.4. This ultrafiltration operation was repeated twice, and the solution was concentrated to 0.05 mL. Protein concentration in the presence of 0.1% (*w*/*v*) SDS was measured using the FluoProdige Assay Kit (OZ Biosciences, San Diego, USA) and was typically 40–50 mg/mL. The membrane protein fraction was stored in 10 μL aliquots at −80 °C until use.

Ligand-receptor complex formation was performed as described above, except that the ligand-binding buffer was changed to 20 mM HEPES-NaOH pH 7.4, 300 mM NaCl, and 0.01% (*w*/*v*) LMNG. SEC-MS was performed as described above, except for the following steps. Typically, 2 μL of sample was injected, and a higher-molecular fraction was separated through a SEC column (Inertsil WP300 Diol, 10 × 1.5 mm) using mobile phase A (100 mM ammonium bicarbonate, 0.001% (*w*/*v*) LMNG, and 2% (*v*/*v*) acetonitrile). The flow rate and column temperature were set to 0.025 mL/min and 5 °C, respectively. After mobile phase B (ultra-pure water/methanol/chloroform/formic acid (39.7:59.6:0.5:0.2, *v*/*v*/*v*/*v*), i.e., the mixing ratio of 160:240:2:1) was mixed post-column at 0.175 mL/min, the eluent was introduced into the mass spectrometer. To avoid contaminating the mass spectrometer, the eluate was introduced into the mass spectrometer only for the higher-molecular fractions with 0.5–2.0 min elution times. The duration of each measurement was 5 min.

## 5. Conclusions

This SEC-MS method allowed us to analyze the interaction of a mixture of medicinal herbal ingredients with a mixture of membrane proteins to identify the two interacting ingredients. As it does not require specialized 2D-LC-MS, this method enables the analysis of interactions between LMW compounds and relatively high-expressed membrane proteins without immobilization or derivatization of the molecules.

## Figures and Tables

**Figure 1 molecules-27-04889-f001:**
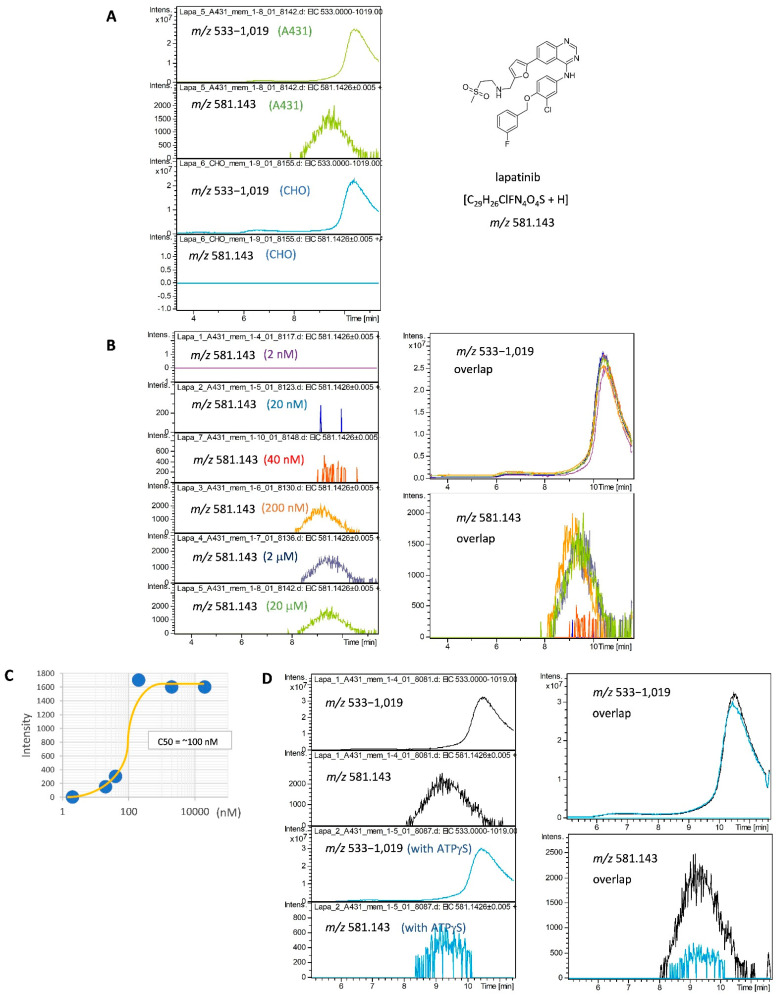
Detection of interaction between the A431 cell membrane protein fraction and EGFR inhibitor, lapatinib. (**A**) Comparison of A431 cells with highly expressed EGFR and CHO cells without EGFR. In total, 20 μM of lapatinib was added to the membrane protein fraction from A431 or CHO cells, followed by injection of the mixture into SEC-MS. The elution profiles of lapatinib binding to its specific target protein were monitored using detection at *m*/*z* 581.143. LMW compounds binding non-specifically to membrane proteins were monitored using detection at *m*/*z* 533−1019. (**B**) Concentration dependence of the lapatinib interaction. In total, 2 nM to 20 μM of lapatinib was added to the membrane protein fraction from A431; then, the interaction between lapatinib and membrane proteins was analyzed. (**C**) Interaction curve between lapatinib and the A431-derived membrane proteins by SEC-MS. The curve showed that the 50% interaction concentration was ~100 nM using the SEC-MS method. (**D**) Effect of ATPγS on the lapatinib interaction. In total, 2 μM of lapatinib was added to the membrane protein fraction from A431, together with or without 600 μM ATPγS, and the interaction between lapatinib and membrane proteins was analyzed.

**Figure 2 molecules-27-04889-f002:**
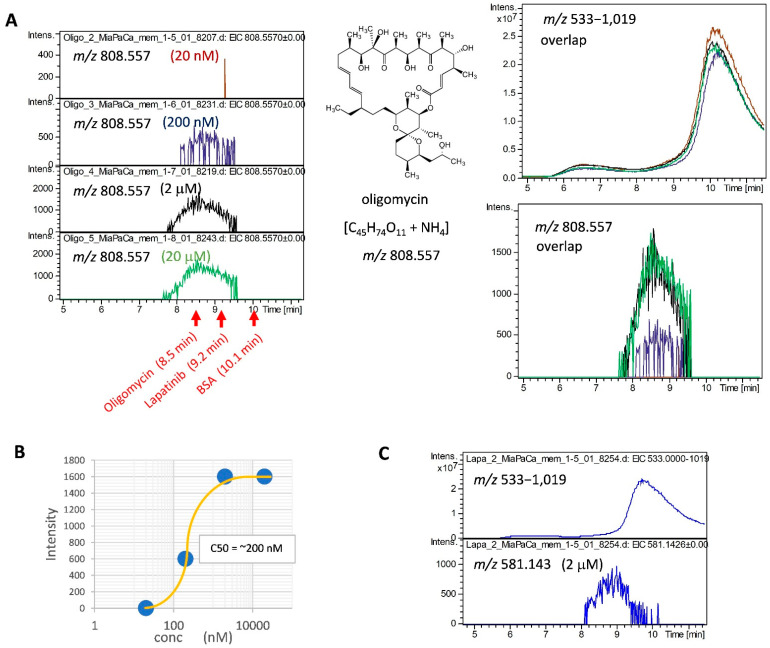
Detection of interaction between the membrane protein fractions from MiaPaCa2 cells and the membrane protein-specific inhibitors. (**A**) Concentration dependence of the ATP synthase inhibitor-interaction. In total, 20 nM to 20 μM of oligomycin was added to the membrane protein fraction from MiaPaCa2; then, the interaction between lapatinib and membrane proteins was analyzed. The elution profiles of oligomycin binding to its specific target protein were monitored using detection at *m*/*z* 808.557. (**B**) Interaction curve between oligomycin and the MiaPaCa2-derived membrane proteins by SEC-MS. The curve showed that the 50% interaction concentration was approximately 200 nM using the SEC-MS method. (**C**) Detection of interaction between lapatinib and the MiaPaCa2-derived membrane proteins by SEC-MS. In total, 2 μM of lapatinib was added to the membrane protein fraction from MiaPaCa2, and the interaction between lapatinib and membrane proteins was analyzed.

**Figure 3 molecules-27-04889-f003:**
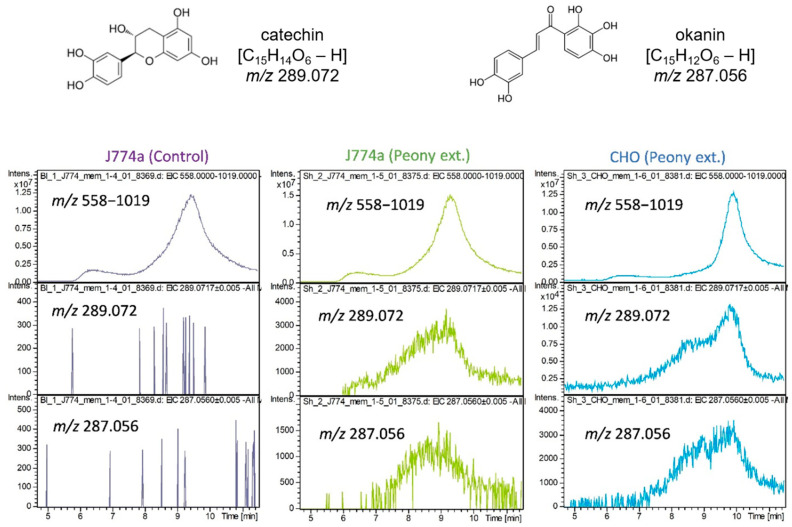
Detection of interactions between membrane proteins and peony extracts. Membrane proteins were extracted from J774a, or CHO cells cultured in a 100 mm dish. The peony or control extract was added to the membrane protein fractions, and the mixture was injected into the SEC-MS. Interactions were analyzed for the top 50 *m*/*z* signals detected in the ingredient analysis of the extract using the APCI-negative detection mode. The results showed interactions between the peony extract catechin (*m*/*z* 289.072) or okanin (*m*/*z* 287.056, presumably assigned) and the proteins extracted from the cells.

**Figure 4 molecules-27-04889-f004:**
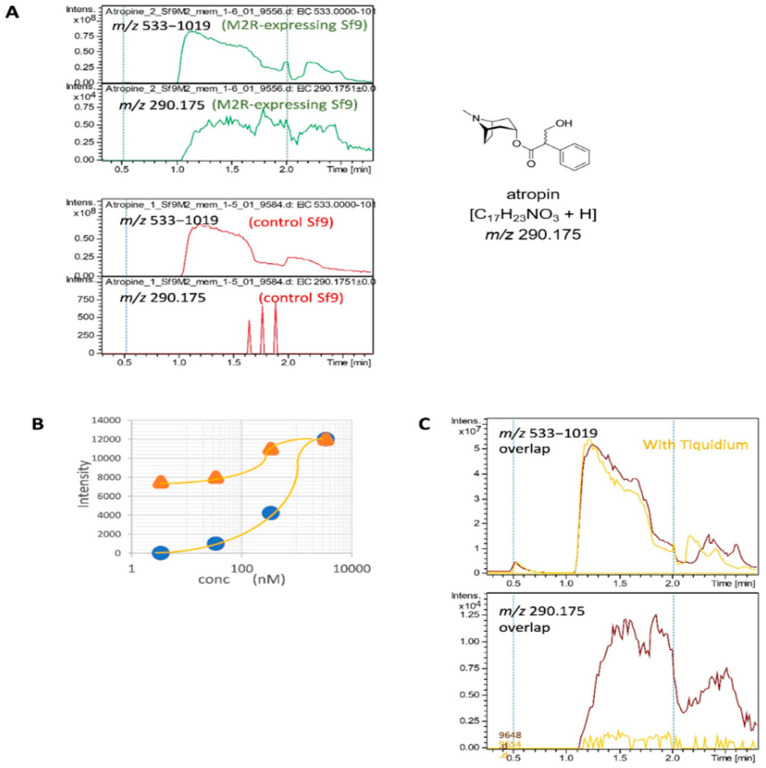
Detection of interaction between the membrane protein fractions from M2R-expression Sf9 cells and the M2R antagonist, atropine. (A) Comparison of Sf9 cells with high expression of M2R and the control Sf9 cells. In total, 340 nM of atropine was added to the membrane protein fraction from Sf9 cells, and the mixture was injected into SEC-MS. The elution profiles of atropine binding to its specific target protein were monitored using detection at *m*/*z* 290.175. LMW compounds binding non-specifically to membrane proteins were monitored using detection at *m*/*z* 533−1019. (B) In-teraction curve between atropine and the M2R-expressing Sf9-derived membrane proteins by SEC-MS. The membrane proteins were solubilized with (triangle) or without (circle) atropine and analyzed. (C) Effect of tiquizium on the atropine interaction. Atropine (3.4 µM) was added to the membrane protein fraction, together with or without 240 μM tiquidium; then, the interaction be-tween atropine and membrane proteins was analyzed.

## Data Availability

Further information and requests for resources and reagents should be directed to the corresponding author (hogiso.tky@gmail.com). The mass spectrometry data have been deposited to Mendeley Data (http://dx.doi.org/10.17632/5sr2ys6tjr.1, published on 27 July 2022). Any additional information required to reanalyze the data reported in this paper is available from the lead contact upon request.

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
