# Peer review of "A Liquid Chromatography-Mass Spectrometry Method to Study the Interaction between Membrane Proteins and Low-Molecular-Weight Compound Mixtures"

_molecules, 2022, doi:10.3390/molecules27154889_

Round 1

Reviewer 1 Report

The present manuscript entitled "A liquid chromatography-mass spectrometry method to study the interaction between membrane proteins and low-molecular-weight compound mixtures" by Hideo Ogiso, Ryoji Suno, Takuya Kobayashi, Kawami Masashi, Mikihisa Takano, and Masaru Ogasawara (molecules-1840314) is written correctly and has a good structure; moreover, it has all the necessary parts. The article is very interesting from an analytical point of view; therefore, it should interest the reader. It is worth highlighting the included section on the limitations of the method. I only have a few questions about the article. The paper meets Molecules' requirements, and I recommend the article for publication in Molecules following the common editing stage. My current decision is a minor revision.

More specific comments and observations are presented below.

1. Please check the literature format if it complies with the journal's requirements. The references at the end of the manuscript are numbered, and no numbers appear in the text. The references on pages 16 and 17 do not comply with the Molecules’ requirements.

2. Page 3, section 2.1. The authors in the second paragraph mention the absence of interference. What can be done in the event of strong interference effects? How would you deal with them? What types of interference effects could occur?

3. Figures (also in SI). Drawings should be visually consistent (taking into account the size). You can delete redundant text, especially the text describing specific measurements. Data can be exported and prepared in another program with well-defined axes and units. Indexes are missing for chemical formulas (also in some places in the text).

4. Page 7. Table S2 is mentioned. There is no such table in SI. Is it a table in a given reference?

5. Table 1. The table is not mentioned in the text. You should add a short introduction to what is there. Apart from the company, you can add the countries of origin to the table.

6. Experimental part. A diagram showing the activities performed would be interesting for the reader. Please consider this point.

7. Page 15. Datum or data?

8. Please add a section with conclusions.

9. Is it possible to validate the method by counting validation parameters?

10. It would be worthwhile to evaluate the method using RGB Additive Color Model to Analytical Method Evaluation or AGREE-Analytical GREEnness Metric Approach.

I hope that the comments presented will help improve the article.

Author Response

Dear Sir,

Thank you for taking time out of your busy schedule to review our manuscript. We have made the following changes according to your comments. I have marked in red letters in the text that was changed according to the reviewer's comments.

Reviewer #1

1.

According to the reviewer’s comment, numbering has been done to the cited references.

2.

The issue to be solved in developing this method was ion suppression, so by optimizing the detergent and post-column solution conditions and ionization with APCI, the signal of the LMW ligand could be detected. Ion suppression was observed even with APCI when the values of the capillary voltage and corona current of the ion source were low, so these were set to values high enough not to cause ion suppression. If ion suppression is observed when measuring some other sample, it may be necessary to change the set values to even higher values. We have not experienced this in this study so far and therefore refrain from describing it in the text.

3.

I am sorry not to understand this comment. Please point out specifically which part of the chemical formula index should be changed. Regarding the chromatogram, we do not have the special tools to process it, so we create the chromatogram figures of the same size as much as possible based on the output diagrams of the DataAnalysis software for LC-MS measurements.

4.

Of the supplementary information, the supplementary tables are Excel files; as it is not possible to upload Excel files, they have been sent as email attachments to the editorial team.

5.

I have changed Table 1 to Table S3. A sentence describing Table S3 has been added to “Materials and Methods” in the text.

6.

A conceptual diagram of the whole operation has been added as a “Graphical Abstract”.

7.

I am very sorry not to understand this comment. Please point out specifically which of the "data" should be changed.

8.

Conclusions section has been added to fit the MOLECULES format.

9.

I am sorry for my lack of knowledge and understanding. What exactly do you mean by “counting validation parameters”? I have never done such an evaluation before and would like to study it in the future.

10.

I have no knowledge of the "RGB Additive Colour Model to Analytical Method Evaluation" or the "AGREE-Analytical GREEnness Metric Approach" and would like to study them in the future. I would be grateful if you could provide me with some references that are easy to understand even for beginners.

Thank you for your consideration. I look forward to hearing from you.

Sincerely,

Hideo Ogiso

Toyama Prefectural Institute for Pharmaceutical Research,

Nakataikouyama, Imizu-shi, Toyama 939-0363, Japan;

E-mail address:  hogiso.tky@gmail.com

Reviewer 2 Report

Comments for the authors:

In this manuscript, the authors developed a convenient and throughput-oriented method for analyzing the interaction between a membrane protein mixture and an LMW compound mixture using novel simple SEC-MS. The method was applied and validated for the analysis of the molecular interactions between specific inhibitors (e.g. lapatinib and oligomycin) and their stable target membrane proteins. The authors provided an example of the application of this method by examining the interaction of an LMW mixture extracted from P. lactiflora with a mixture of membrane proteins from two different types of cells, and evaluated the mass chromatograms corresponding to the interactions in this measurement. Also, the authors tried to determine whether interactions could be detected for unstable target GPCRs using a modified SEC-MS method and discussed the result comprehensively and honestly by pointing out the limitations of the method.

The experiments were designed well with proper controls. The authors have a strong track record in the methods they have described. The manuscript is incredibly information-rich, and I provided some questions and suggestions here for the authors’ consideration:

1. Introduction: “Biological macromolecules interact with other biomolecules or bioactive low-molecular-weight (LMW) compounds via noncovalent interactions …”

Rewrite this sentence. Covalent interactions between macromolecules and LMW compounds are also very common, such as b-lactams and penicillin-binding proteins. Change the sentence to make sure it does not exclude or ignore other interaction types.

2. Introduction: “Since many membrane proteins, such as receptor tyrosine kinases, ion pumps, and G-protein coupled receptors (GPCRs), are target molecules for drugs …”

Change “are target molecules for drugs” to “have been reported as target receptors for drugs”

3. Introduction: “Another method is size-exclusion chromatography (SEC), which separates proteins complexed with their low molecule ligands by size and is used to determine …”

Change “which separates” to “which can separate”;

Change “is used to determine” to “has been used to determine”.

4. Figure 3: Label the catechin and okanin with their m/z values on the figure. This would reads more clear to the reader.

5. 2.4 part: “Two patterns were analyzed using membrane protein fractions prepared with or without atropine added during solubilization (Figure 4A, B; Figure S2).”

Explain this with more details, it is a little bit confusing.

6. Discussion: “The fact that the 50% binding concentration of 700 nM in this study was more than 200-fold lower than the IC50 …”

Double check about this sentence. It seems “20-fold higher” to the reviewer.

7. Discussion: “Thus, further improvements are need GPCR analyses.”

Change to Thus, further improvements are needed for GPCR analyses.”

Author Response

Dear Sir,

Thank you for taking time out of your busy schedule to review our manuscript. We have made the following changes according to your comments. I have marked in red letters in the text that was changed according to the reviewer's comments.

Reviewer #2

1.

According to the reviewer’s comment, I have additionally described “covalent bonds” in the introduction section, and “noncovalent” in the section of limitation of the study.

2.

According to the reviewer’s comment, I have corrected the expression.

3.

According to the reviewer’s comment, I have corrected the expression.

4.

I have added the m/z values in Figure 3 and others.

5.

"To stabilize the structure of GPCRs, they are generally solubilized and purified with addition of ligands. In the present study, " have been added.

6.

I have corrected to "200-fold higher".

7.

According to the reviewer’s comment, I have corrected the expression.

Thank you for your consideration. I look forward to hearing from you.

Sincerely,

Hideo Ogiso

Toyama Prefectural Institute for Pharmaceutical Research,

Nakataikouyama, Imizu-shi, Toyama 939-0363, Japan;

E-mail address:  hogiso.tky@gmail.com